# Estimation of Potato Yield Using Satellite Data at a Municipal Level: A Machine Learning Approach

**Pablo Salvador \*, Diego Gómez** **, Julia Sanz** **and José Luis Casanova**

Remote Sensing Laboratory (LATUV), University of Valladolid, Paseo de Belen 11, 47011 Valladolid, Spain; diego@latuv.uva.es (D.G.); julia@latuv.uva.es (J.S.); jois@latuv.uva.es (J.L.C.)
* Correspondence: pablo@latuv.uva.es; Tel.: +34-983-423952

**Abstract:** Crop growth modeling and yield forecasting are essential to improve food security policies worldwide. To estimate potato (*Solanum tubersum L.*) yield over Mexico at a municipal level, we used meteorological data provided by the ERA5 (ECMWF Re-Analysis) dataset developed by the Copernicus Climate Change Service, satellite imagery from the TERRA platform, and field information. Five different machine learning algorithms were used to build the models: random forest (rf), support vector machine linear (svmL), support vector machine polynomial (svmP), support vector machine radial (svmR), and general linear model (glm). The optimized models were tested using independent data (2017 and 2018) not used in the training and optimization phase (2004–2016). In terms of percent root mean squared error (%RMSE), the best results were obtained by the rf algorithm in the winter cycle using variables from the first three months of the cycle ($R^2 = 0.757$ and %RMSE = 18.9). For the summer cycle, the best performing model was the svmP which used the first five months of the cycle as variables ($R^2 = 0.858$ and %RMSE = 14.9). Our results indicated that adding predictor variables of the last two months before the harvest did not significantly improved model performances. These results demonstrate that our models can predict potato yield by analyzing the yield of the previous year, the general conditions of NDVI, meteorology, and information related to the irrigation system at a municipal level.

**Keywords:** machine learning; meteorological data; municipal level; potato yield; satellite imagery

## 1. Introduction

Potato (*Solanum tuberosum L.*) is a crop that originated from the Andean region [1]. It is the most important non-grain crop worldwide [2] and the fourth-largest food crop after rice, wheat, and maize [3,4]. Potato produces more nutritious food, quicker, on less land and in harsher climates than any other major crop [2]. It also plays a key role in food security strategies being grown and consumed in underprivileged areas [5]. In this sense, millions of farmers rely on potatoes as a source of food and economical support. Potato global trends indicate that developed countries are gradually reducing their potato production, whereas developing countries are increasing it [5,6].

Mexican potato production increased with respect to global production until approximately the year 2000, then its production has remained stable to date [7]. In general, 8700 producers are dedicated to the cultivation of potatoes in Mexico. Approximately 77800 families directly depend on this crop, generating 17500 direct jobs and more than 50 thousand indirect jobs [8]. The latter report noted that potatoes are grown over 23 states of the Mexican republic. By states, Sonora is the main producer with 24.5% of the total, followed by Sinaloa (17.0%), Puebla (9.8%), and Veracruz (8.3%). According to its end use, 29.0% is produced for industrial purposes, 56.0% for fresh consumption, and 15.0% for seed production. Concerning the seasonality of the crop, two main growing periods can be identified: autumn–winter and spring–summer. Autumn–winter periods generally present

yields around 25 ton/ha for irrigated fields and 14 ton/ha for rain-fed. On the contrary, spring–summer periods usually have superior yields for both irrigation (30 ton/ha) and rain-fed (17 ton/ha). Haverkort and Struik [6] studied the influence of meteorological conditions over potato crop development. Air temperature may affect canopy development, tuber bulking, and growth cycle duration, even outside ranges of cold or heat stress. Other studies, such as Timlin et al. [9], highlighted the role of radiation as conditioning factor of potato total yield. The amount of dry matter produced per unit of light intercepted was found to be approximately 2.62 g m$^{-2}$ MJ$^{-1}$ photosynthetically active radiation (PAR) assuming 12 plants per m$^2$ in the chamber and using the measured PAR of 10.2 MJ for the day (47 mol m$^{-2}$ d$^{-1}$). According to FAO (2009), potato yields in the developing world range from 10 to 15 ton/ha, less than half of the average yields achieved by farmers in Western Europe and North America [2].

Crop modeling has greatly evolved in the last four decades, in part facilitated by satellite remote sensing, providing information in large or inaccessible areas in an operative manner. Monteith et al. [10,11] identified the importance of measuring the total amount of photosynthetically active radiation (APAR) (400–700 nm) absorbed by a canopy throughout the growing season to estimate crop yields. Therefore, satellite remote sensing offers the possibility to identify variations in plant physiology through changes in APAR [12]. Given the limitations of this parameter [13], many authors opted to use the fraction of absorbed photosynthetically active radiation (fAPAR) since it plays a critical role in the energy balance of ecosystems and in the estimation of the carbon balance [14]. Thus, fAPAR has been directly associated with photosynthesis and the link between ecosystem function and structure [15]. Aiming to obtain the best approach to derive fAPAR from satellite imagery, some authors suggested the use of the simple ratio (SR) between red (RED) and near infrared (NIR) reflectances over vegetated areas [16,17]. While, in contrast, others recommended the use of normalized difference vegetation index (NDVI) as a fairer approach [18,19]. NDVI is one of the most widely used vegetation indices to evaluate vegetation status and crop yield estimation [20–23].

Nevertheless, its success strongly depends on the type of crop target [22]. The combination of NDVI and climatic data may overcome some of the shortcoming or uncertainties of using only vegetation indices to estimate crop yields [24]. Remote-sensing-based methods to model crop yields require the processing of large amounts of data from different products and sources. As a consequence, machine learning (ML) techniques are increasing in popularity due to their capability to process a large number of inputs and handle non-linear tasks [25].

The overarching objective of the current study is to evaluate the performance of well-established non-linear machine learning algorithms to estimate the potato yield prior the harvesting period at municipal level in Mexico. This methodology can be of great interest to national or municipal authorities in order to improve management and food security policies.

## 2. Materials and Methods

### 2.1. Study Area

The study area is located in Mexico. This country has over 124 million people and covers a total area of 1,964,375 km$^2$, with its territory divided into 32 estates and 2458 municipalities. Given its vast extension, Mexico is a country with a great climatic diversity with two distinct areas separated by the Tropic of Cancer: the tropical zone and the temperate zone (Figure 1). According to Köppen–Geiger classification, the most frequent climate is hot semi-arid climate (BSh) and hot desert climate (BWh) [26]. These climatic conditions allow farmers to exploit the two cycles of potato, one during spring–summer and the other during autumn–winter.

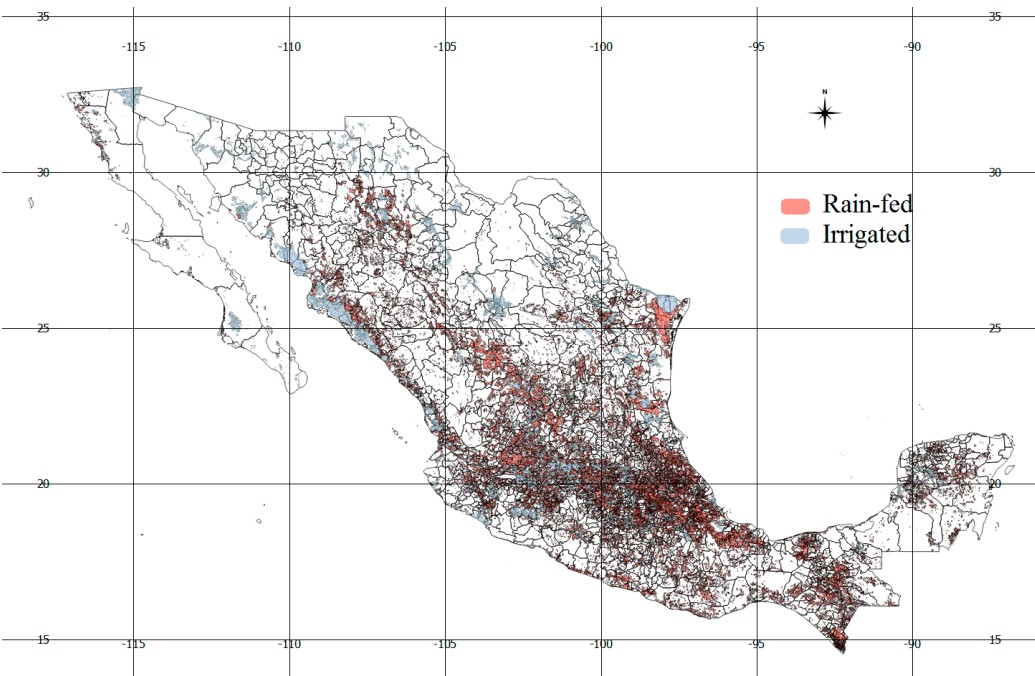

**Figure 1.** Study area with the locations of irrigated and rain-fed crops in Mexico.

## 2.2. Materials

### 2.2.1. In Situ Data

The land cover map v.6.0 (1:25:000) from Instituto Nacional de Estadística y Geografía de México (INEGI) was used to identify those polygons categorized as crops to extract a representative vegetation vigor at a municipal level [27]. Its main advantage is the capacity to differentiate between irrigated and rain-fed crops. In Mexico, there is an approximate area of 34 Mha dedicated to agricultural use which represents 17% of the total area (23.7Mha rain-fed, 10.4 Mha irrigated). The crop yield data was downloaded from the Servicio de Información Agroalimentaria y Pesquera (SIAP) of the Mexican government (http://infosiap.siap.gob.mx/gobmx/datosAbiertos.php). This data comprises information for each year and municipality for different crops such as sown and harvested areas, production volume, total value, irrigated or rain-fed system, seasonality (spring–summer or autumn–winter), or crop yield since 1980. The boundaries of Mexican municipalities were downloaded from the webpage https://www.diva-gis.org/Data, site of the freely available DIVA-GIS software [28].

### 2.2.2. Remote Sensing Data

We used the MOD13Q1 product to represent the vegetation vigor of each Mexican municipality. This product compiles the NDVI (obtained by equation 1) of MODIS (Moderate Resolution Imaging Spectroradiometer) [29].

$$NDVI = (NIR-RED)/(NIR + RED) \qquad (1)$$

The MODIS sensor, on board the Terra platform from the EOS constellation (Earth Observing System), has 36 channels ranging in wavelength from 0.4 to 14.4 μm. The spatial resolution varies from 250 m to 1 km. More information about technical aspects of MODIS can be found in Ranson et al. [30]. The MOD13Q1 is a 16-day global composite of NDVI with a spatial resolution of 250 m, downloaded from the EarthData website for the following tiles: h09v06, h08v06, h08v07, h07v06, h09v07, and h08v05 [31]. We calculated a monthly composite image with the maximum value of NDVI. To do that, a mosaic image was created every 16 days by joining the corresponding six tiles and then, the maximum value per month was selected in the final monthly composite for each pixel

mitigating, this way, the cloud cover related problems. Although the enhanced vegetation index (EVI) is also computed inside MOD13Q1 product, it has been dismissed because of its high correlation with NDVI [32,33].

### 2.2.3. Meteorological Data

The meteorological data was retrieved from ERA5 (ECMWF Re-Analysis) developed by the Copernicus Climate Change Service (C3S). ERA5 is the fifth generation ECMWF (European Centre for Medium-Range Weather Forecasts) reanalysis for the global climate and weather of the last four to seven decades under a free of charge license, worldwide, non-exclusive, royalty free, and perpetual. Data since 1979 is available. The reanalysis combines model data with world-wide observations into a globally complete and consistent dataset using model physics, core dynamics and data assimilation [34]. The input variables of the model (Table 1) were downloaded in a monthly mean basis, with a horizontal resolution of 0.25 degrees in a global coverage, from the following webpage: https://cds.climate. copernicus.eu/cdsapp#!/dataset/reanalysis-era5-single-levels-monthly-means?tab=overview.

**Table 1.** Meteorological variables downloaded from ERA5 (ECMWF Re-Analysis) developed by the Copernicus Climate Change Service (C3S).

| Name | Short Name | Units |
| --- | --- | --- |
| Evaporation | e | m of water equivalent |
| Leaf area index, high vegetation | lai_hv | $m^2/m^2$ |
| Leaf area index, low vegetation | lai_lv | $m^2/m^2$ |
| Potential evaporation | pev | m |
| Skin temperature | skt | K |
| Surface net solar radiation | ssr | $J/m^2$ |
| Surface pressure | stl1 | Pa |
| Volumetric soil water layer 1 | swvl1 | $m^3/m^3$ |
| 2-meter temperature | t2m | K |
| Total cloud cover | tcc | Times one |
| Total precipitation | tp | m |

Figure 2 shows, as an example, the temporal evolution of the NDVI and meteorological variables in Comondú municipality, Baja California, from October 2017 to September 2018. This state corresponds to irrigated farmer system for both cycles. The yield production for this year was 38 ton/ha for the winter cycle and 31 ton/ha for the summer cycle. Aiming to represent all the variables in one figure, skt and t2m have been divided by 1000, stl1 was measured in MPa, and tp, e, pev, and tp were measured in cm.

### 2.3. Methods

#### 2.3.1. Data Preparation

The yield information was extracted at municipal level for the time period 2003–2018. The variable predictors were the farmer system (irrigated or rain-fed crop information derived from SIAP database), the yield production during the year before (ybYield) which represents the simplest prediction, NDVI and meteorological variables (Table 1). To generate these predictors, we used the RSI's software ENVI/IDL [35,36]. We extracted the mean NDVI value per month and municipality from those pixels classified as crop over six months (between sowing and harvest). It is worth noting that MOD13Q1 is a 16-day global composite, therefore while some months had only one value of NDVI per pixel, others had two values of NDVI per pixel. Thus, when a pixel had two values within a month, the maximum value was selected. The months from October to March were included in the winter cycle, and from April to September in the summer cycle. Given the low resolution of the meteorological data,

these predictors were retrieved at municipal level, on a monthly basis, between the sown and harvest time for the six months before the harvesting. Hence, the mean value of each variable included in Table 1 was retrieved. Therefore, the dataset compiled one yield record per municipality and year, with its corresponding monthly mean NDVI and meteorological variables of the six months prior harvesting. Data samples with missing values were removed from the dataset. In addition, field data with yields lower than 10 ton/ha were considered as outliers and removed. The final dataset comprised 838 samples.

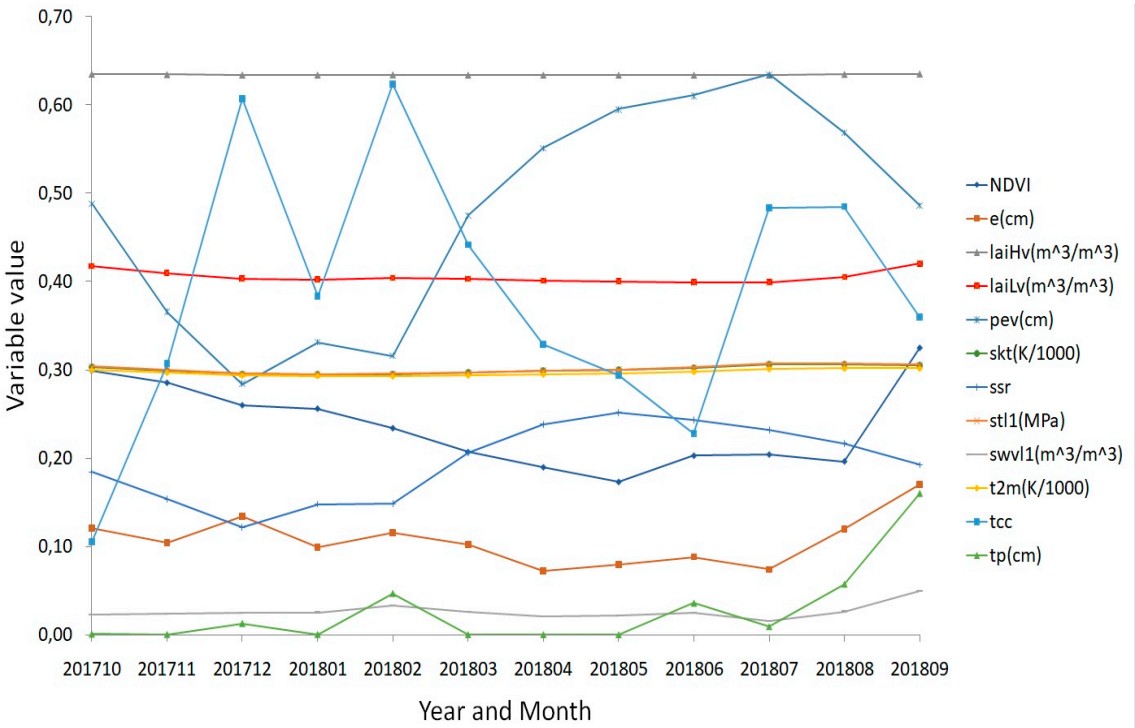

**Figure 2.** Variables evolution from October 2017 to September 2018 in the municipality of Comondú in the state of Baja California.

### 2.3.2. Machine Learning Modeling

During the modeling process, we used R software [37] and the caret package [38]. We set up six different scenarios based on which months were used (only predictors from the sixth month before harvest, then sixth and fifth month before harvest, and so on until all the available predictors from the six months before harvest were included). In addition, each model was built with five different ML algorithms: random forest (rf) [39], support vector machine linear (svmL) [40], support vector machine polynomial (svmP) [40], support vector machine radial (svmR) [41], and general linear model (glm) [42].

The dataset was split into a training and validation set (85%, 712 samples), comprising data from 2004 to 2016; and an independent test set (15%, 126 samples) with data from 2017 and 2018. Thus, the percentage of data assigned to each subset was conditioned by the amount of samples available from 2017 and 2018. This method intends to simulate the performance of the models in a more 'real-world' scenario, finding a compromise between number of samples included in the test set and selecting only the most recent data. We used the bootstrapping resampling method (25 repetitions) to train the model and optimize the hyperparameters.

In order to assess the variability of this methodology, we replicated the previous approach over 50 iterations, but this time randomly selecting 90% of the data at each iteration. The best performing models were then evaluated with the latest data records (2017 and 2018) to assess their capacity in

terms of operability (simulating a real-case scenario). The final and operative model would therefore be built with the entire dataset (2004–2018).

## 3. Results

The model performance of each ML algorithm per scenario and cycle using the hold-out data (2017–2018) using the bootstrapping method with 25 iterations is shown in Table 2. The accuracy of the models were presented in terms of coefficient of determination ($R^2$) and percent root mean squared error (%RMSE).

**Table 2.** Model performance per algorithm, cycle, and months used to create variables from the harvest date. The test dataset corresponds to the independent hold-out set (2017 and 2018).

| Cycle | Scenario | rf | | svmR | | svmL | | svmP | | glm | |
|---|---|---|---|---|---|---|---|---|---|---|---|
| | | $R^2$ | RMSE | $R^2$ | RMSE | $R^2$ | RMSE | $R^2$ | RMSE | $R^2$ | RMSE |
| Autumn–winter | 1 | 0.750 | 18.0 | 0.732 | 22.1 | 0.699 | 20.6 | 0.691 | 22.3 | 0.700 | 19.7 |
| | 2 | 0.763 | 19.0 | 0.752 | 22.4 | 0.701 | 21.6 | 0.720 | 23.4 | 0.668 | 23.6 |
| | 3 | 0.764 | 19.0 | 0.750 | 22.9 | 0.691 | 23.0 | 0.717 | 23.3 | 0.641 | 26.2 |
| | 4 | 0.763 | 18.9 | 0.723 | 22.4 | 0.688 | 23.2 | 0.716 | 24.2 | 0.652 | 27.8 |
| | 5 | 0.760 | 19.1 | 0.714 | 22.4 | 0.670 | 21.4 | 0.701 | 24.3 | 0.674 | 23.5 |
| | 6 | 0.760 | 19.7 | 0.704 | 22.5 | 0.659 | 26.2 | 0.694 | 24.5 | 0.248 | 31.2 |
| Spring–summer | 1 | 0.809 | 17.6 | 0.848 | 15.3 | 0.876 | 14.4 | 0.837 | 15.7 | 0.841 | 16.6 |
| | 2 | 0.824 | 16.7 | 0.848 | 15.6 | 0.873 | 14.9 | 0.816 | 17.4 | 0.855 | 16.2 |
| | 3 | 0.826 | 17.0 | 0.855 | 14.8 | 0.867 | 15.3 | 0.870 | 14.3 | 0.855 | 16.5 |
| | 4 | 0.845 | 16.3 | 0.850 | 14.8 | 0.865 | 15.1 | 0.868 | 14.4 | 0.846 | 16.1 |
| | 5 | 0.843 | 16.3 | 0.842 | 15.6 | 0.847 | 16.5 | 0.854 | 15.4 | 0.814 | 17.6 |
| | 6 | 0.835 | 16.6 | 0.791 | 18.0 | 0.838 | 16.2 | 0.842 | 15.9 | 0.790 | 17.9 |

Table 3 shows the model performance of each ML algorithm per scenario and cycle using the bootstrapping method (25 repetitions) across 50 iterations, expressed as the mean and standard deviation of $R^2$ and %RMSE scores. This method allows us to compare the variance of the model resulting from each iteration (50 in total), and thereby the overall influence of the random sampling in the training phase.

In terms of %RMSE, the best ML algorithm during the winter cycle was the rf using the predictors corresponding to the 3-month scenario. In the summer cycle, the best results were obtained by svmP algorithm when using the predictors corresponding to the 5-month scenario (Table 3). In the winter cycle, the models after the 3-month scenario did not considerably improve $R^2$ and %RMSE scores, whereas model performances increased until the 5-month scenario in the summer cycle.

The crop cycle variable affected differently depending on the ML model used. In the summer cycle, the best svmP performance for the 5-month scenario ($R^2 = 0.858$) was substantially better than the rf performance ($R^2 = 0.757$) in winter for the 3-month scenario. Similar results were observed for other ML models and their best scenario: svmR ($R^2 = 0.731 - 0.837$), svmL ($R^2 = 0.692 - 0.860$), and glm= ($R^2 = 0.612 - 0.834$). Given the difference performance observed between summer and winter cycles, we did not build a unique model that would comprise data from both cycles. Some differences were also observed depending on whether the crops were irrigated or non-irrigated. In general, the models were better in the irrigated crops (%RMSE summer =8.9, %RMSE winter = 17.0) than in non-irrigated ones (%RMSE summer =22.5, %RMSE winter = 22.8).

As stated in the methodology section, we trained and optimized the models using the bootstrap method (25 repetitions) with 50 iterations. The best number of variables tried at each split (mtry) for the rf model (winter cycle, 3-month scenario) was 20. This hyper-parameter gave the best results in terms of RMSE for 47 out of 50 iterations. For the svmP model (summer cycle, 5-month scenario) the best hyper-parameters were degree = 3, scale = 0.01 and offset = 1 (49 out of 50 iterations). The support vector number was 306.

**Table 3.** Model performance of each algorithm, cycle, and scenario across 50 iterations, expressed as mean values of $R^2$ and %RMSE, and their corresponding standard deviations. The test dataset corresponds to the independent hold-out set (2017 and 2018).

| | Cycle | Scenario | rf | | svmR | | svmL | | svmP | | glm | |
|---|---|---|---|---|---|---|---|---|---|---|---|---|
| | | | $R^2$ | RMSE | $R^2$ | RMSE | $R^2$ | RMSE | $R^2$ | RMSE | $R^2$ | RMSE |
| **Mean values** | Autumn–winter | 1 | 0.764 | 19.0 | 0.733 | 22.1 | 0.689 | 22.3 | 0.717 | 23.2 | 0.612 | 24.9 |
| | | 2 | 0.755 | 19.1 | 0.729 | 22.5 | 0.687 | 22.3 | 0.713 | 23.4 | 0.596 | 25.6 |
| | | 3 | **0.757** | **18.9** | 0.729 | 22.2 | 0.692 | 22.2 | 0.711 | 23.2 | 0.584 | 25.1 |
| | | 4 | 0.751 | 19.4 | 0.726 | 22.8 | 0.682 | 22.6 | 0.706 | 23.6 | 0.594 | 25.2 |
| | | 5 | 0.762 | 19.3 | 0.731 | 22.6 | 0.688 | 22.7 | 0.712 | 23.6 | 0.589 | 25.7 |
| | | 6 | 0.760 | 19.0 | 0.727 | 22.3 | 0.690 | 22.5 | 0.707 | 23.4 | 0.598 | 24.9 |
| | Spring–summer | 1 | 0.839 | 16.3 | 0.833 | 16.0 | 0.863 | 15.3 | 0.857 | 15.1 | 0.830 | 16.9 |
| | | 2 | 0.839 | 16.3 | 0.837 | 15.9 | 0.857 | 15.7 | 0.859 | 15.1 | 0.830 | 17.0 |
| | | 3 | 0.832 | 16.6 | 0.831 | 16.1 | 0.856 | 15.8 | 0.853 | 15.3 | 0.829 | 17.1 |
| | | 4 | 0.837 | 16.5 | 0.835 | 16.0 | 0.860 | 15.4 | 0.854 | 15.2 | 0.833 | 16.8 |
| | | 5 | 0.839 | 16.0 | 0.834 | 15.8 | 0.859 | 15.4 | **0.858** | **14.9** | 0.834 | 16.6 |
| | | 6 | 0.839 | 16.4 | 0.836 | 16.0 | 0.860 | 15.7 | 0.862 | 15.0 | 0.828 | 17.2 |
| **Standard deviation** | Autumn–winter | 1 | 0.025 | 1.5 | 0.026 | 1.5 | 0.040 | 2.9 | 0.035 | 1.6 | 0.144 | 4.4 |
| | | 2 | 0.031 | 1.4 | 0.031 | 1.2 | 0.035 | 2.4 | 0.035 | 1.5 | 0.154 | 3.9 |
| | | 3 | 0.026 | 1.6 | 0.026 | 1.4 | 0.038 | 2.7 | 0.025 | 1.3 | 0.185 | 5.1 |
| | | 4 | 0.019 | 1.1 | 0.030 | 1.4 | 0.025 | 2.2 | 0.034 | 1.4 | 0.163 | 3.8 |
| | | 5 | 0.037 | 1.7 | 0.032 | 1.5 | 0.032 | 2.3 | 0.027 | 1.6 | 0.176 | 5.1 |
| | | 6 | 0.029 | 1.2 | 0.022 | 0.8 | 0.034 | 2.8 | 0.018 | 1.0 | 0.161 | 4.1 |
| | Spring–summer | 1 | 0.012 | 0.7 | 0.023 | 1.1 | 0.015 | 0.8 | 0.016 | 0.9 | 0.032 | 1.2 |
| | | 2 | 0.015 | 0.8 | 0.021 | 1.1 | 0.018 | 1.0 | 0.015 | 0.8 | 0.029 | 1.2 |
| | | 3 | 0.017 | 0.8 | 0.020 | 1.0 | 0.017 | 1.0 | 0.016 | 0.8 | 0.028 | 1.0 |
| | | 4 | 0.014 | 0.7 | 0.022 | 1.1 | 0.017 | 0.9 | 0.018 | 0.9 | 0.027 | 0.9 |
| | | 5 | 0.015 | 0.6 | 0.021 | 1.0 | 0.017 | 0.9 | 0.014 | 0.7 | 0.030 | 1.2 |
| | | 6 | 0.017 | 0.8 | 0.021 | 1.2 | 0.017 | 1.0 | 0.016 | 0.9 | 0.032 | 1.2 |

We evaluated the statistical significance of each variable in the best performing models and cycle. The variable importance was calculated using the varImp function included in the caret package [38] and scaled to 100 for the sake of visibility. Figure 3 and 4 show the 20 most important variables for the rf model using the 3-month scenario (winter cycle), and svmP model using the 5-month scenario (summer cycle).

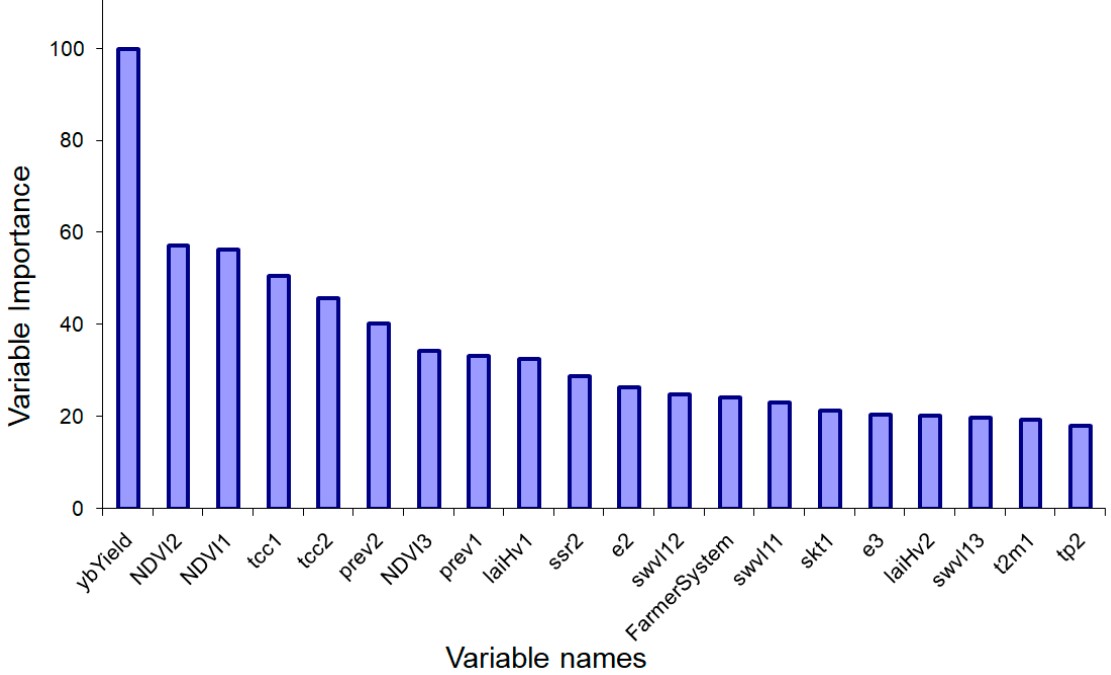

**Figure 3.** Variable importance of the rf model, 3-month scenario and winter cycle using bootstrapping method (25 repetitions) and one iteration.

The variable importance shows in Figures 3 and 4 is only representative and depend on the random selection of the data. In order to perform a deeper analysis of the variable importance, the results of the 50 iterations have been calculated. In the winter cycle, the most important variable for the best performing model (rf, 3-month scenario) was the ybYield in 48 out of 50 iterations. In decreasing order of importance, NDVI1 was also a very relevant variable. It was at the top of the rank (first to third variable importance) in 25 out of 50 iterations. While NDVI2 was the second or third most important variable in 37 out of 50 iterations. In the summer cycle, the most important variables for the svmP model (5-month scenario) concur with the results presented in Figure 4. The most important variables were laiLv 1-5, laiHv 1-5, ybYield, farmer system, NDVI4, and NDVI5 in 47 out of 50 iterations. Although ybYield was in the eleventh place, its level of importance is similar to laiLv or laiHv in terms of % importance. We analyzed the prediction capacity of the ybYield to obtain a baseline-value of the predicted yield. This is one of the simplest models we can get (only one predictor), it is easy to implement and can be used to test and compare our model results. The baseline-value obtained model performances of $R^2$ =0.767 and %RMSE= 20.5 in the summer cycle, while $R^2$ = 0.515 and %RMSE = 25.0 in the winter cycle. Comparing with model results provided in Tables 2 and 3, we thus prove the important information added by the remote sensing and meteorological variables. To further inspect and verify the predictive capacity of the latter variables (remote sensing data and meteorology), we dropped off the ybYield variable from the rf model (3-month scenario, winter cycle) and svmP model (5-month scenario) with %RMSE scores of 19.9 and 23.3, respectively.

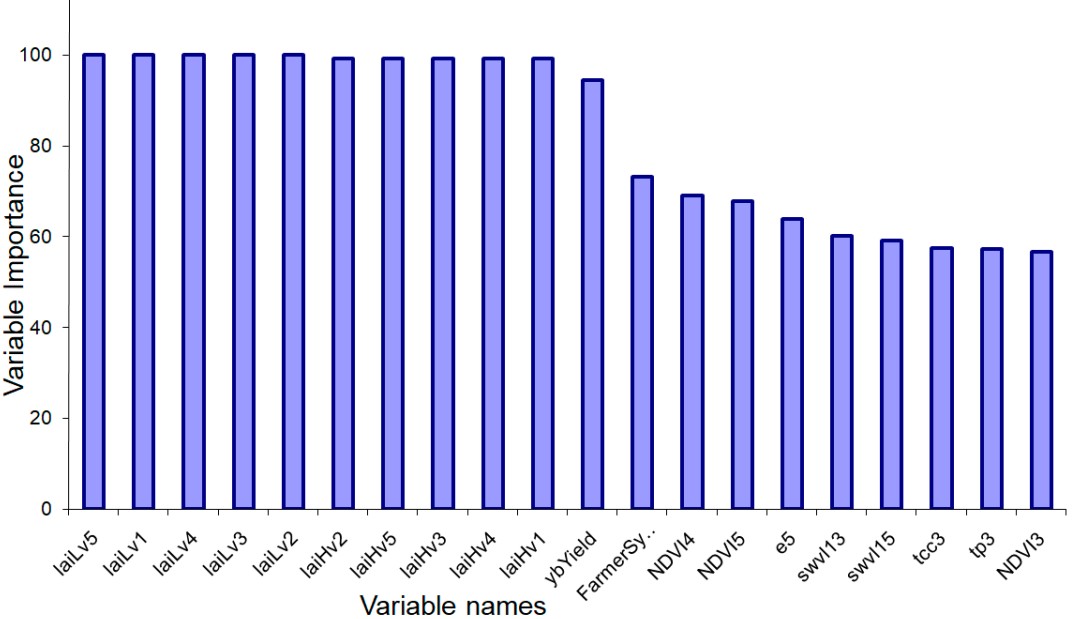

**Figure 4.** Variable importance of the svmP model, 5-month scenario, and summer cycle using bootstrapping method (25 repetitions) and one iteration.

Figure 5 shows the prediction yield versus the actual measured yield for the years 2017 and 2018 (hold-out dataset) for the best performing models at each cycle. A regression has been calculated for this scatter plot, and it resulted in a slope of 0.711 for winter cycle and 0.878 for summer cycle. The intercepts at 4.711 (winter cycle) and 1.482 (summer cycle) indicate the similarity of the linear relation with the 1:1: line.

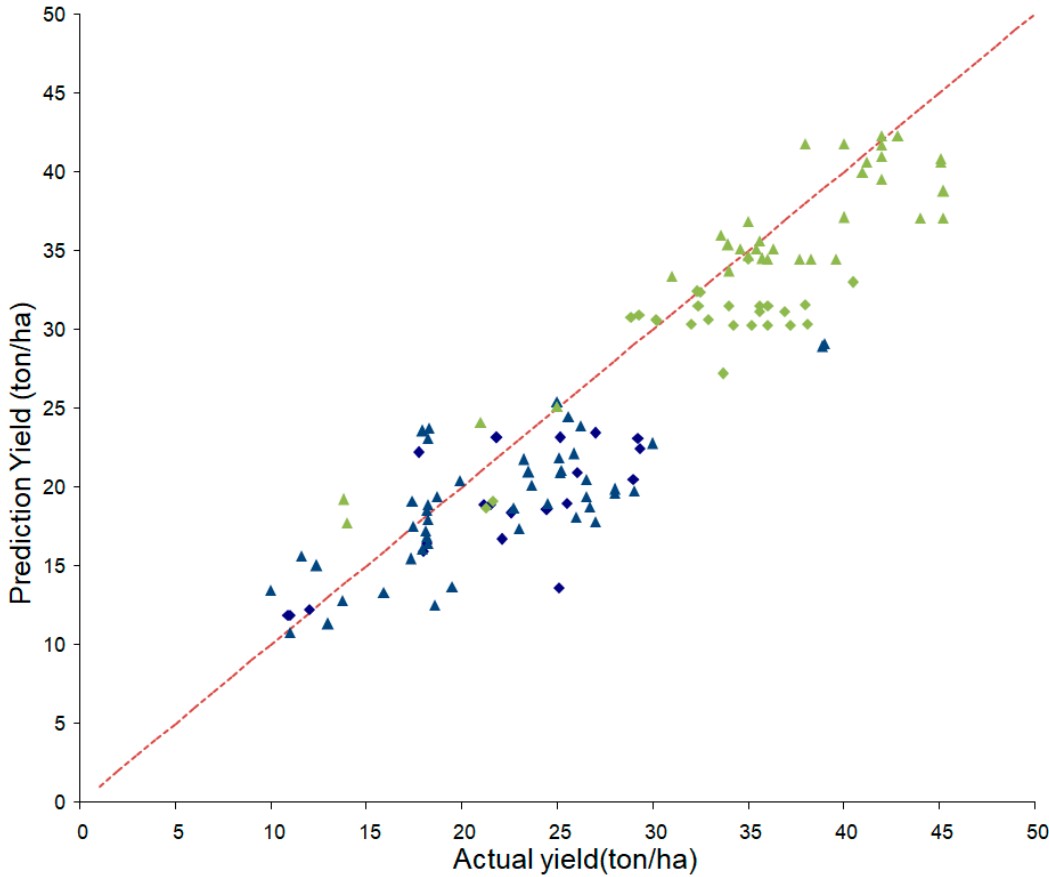

**Figure 5.** Comparison between predicted and actual yields in the independent dataset (2017–2018). Navy blue color represents rain-fed crops, green color represents irrigated crops, diamond-shaped objects represent prediction and actual yield for the rf model (3-month scenario, winter cycle) and triangle-shape objects represent prediction and actual yield for the svmP model (5-month scenario, summer cycle).

## 4. Discussion

Monitoring crop condition and production estimates at state and regional level has been of great interest during the last decades for public authorities and decision makers [43]. In this sense, we built several ML models to estimate potato yields at municipal level in Mexico. One of the most obvious limitations in ML is the lack of data and it can be usually found in the agricultural sector, in which high quality data at finer spatial resolutions sometimes does not exist or it is very limited [44,45]. Given the lack of data at field scale in our study area (phenological information, sown and harvest date), our models used the monthly mean NDVI of the agricultural irrigated or rain-fed fields (depending on the sample), the ybYield and the weather conditions in each municipality as predictive variables. The NDVI used in this work comes from all crops, regardless they are potato cultivation or not. The specific location of the potato crops would provide more precise information to the models with respect to their vigorous state. In addition, the inclusion of other vegetation indexes with the same or higher spatial resolution [46], and not correlated with the information provided by the NDVI, could improve model performances.

The svmP model obtained the best performance in summer cycles using 5-month scenario to create model predictors ($R^2$ = 0.858 and %RMSE = 14.9). Nevertheless, the inclusion of data from the last months before harvest did not substantially improve yield predictions. The NDVI has proved to be an informative indicator of the general agricultural conditions at municipal level during the winter cycle, although the most important variable was the ybYield (Figure 3). Despite NDVI not being retrieved at a

field level, these results reaffirm the potentiality of this vegetation index to characterize the eco-climatic conditions of an area [23,47–50]. In the summer cycle, the most important variables were layhv, lailv, and ybYield (Figure 4). In both models, rf and svmP, the ybYield variable played an important role to predict next year yield; however, our results suggest that its effects are modulated by NDVI and ERA5 data. The prediction capacity of the models using only ybYield ($R^2$ = 0.515/0.767 and %RMSE = 25.0/20.5) were much lower than the results obtained using all the variables (Tables 2–3). Thus, the ybYield variable can be considered as a very informative variable in those cases when standalone remote sensing is not sufficient to reach the desired accuracy. The irrigated and rain-fed systems play different role during both cycles as the rainfall pattern varies substantially in both periods. From 2014 to 2018, the accumulated rain during the summer cycle was about 74% of the total precipitation in a year in average [51]. During the summer cycle, it plays an important role (73% of importance) while during the winter cycle it is not as relevant (24%).

Regardless of the applied ML algorithm, the models performed far better when potato yields were estimated during summer rather than in winter cycles (Tables 2–3). Mean NDVI lailv and laihv values may just represent well potato cultivars during the summer season since the overall value of the evaluated crops is mixed (agricultural crops according to the land cover map). Our results go in accordance with previous studies which estimated potato yield using satellite imagery, but this comparison should be taken carefully as these results were obtained at field scale, and not at municipal level. For instance, Newton et al. [52] addressed the relation between NDVI from Landsat 5-7-8 and potato crop yield using the maximum and mean NDVI values at field-scale obtaining $R^2$ = 0.35 and $R^2$ = 0.81, respectively. Bala and Islam [53] developed a potato yield model using NDVI from MODIS with an estimate error of 15 %RMSE, obtained from data of 50 fields. Our study used 838 field samples to build the models, which improves the robustness of previous approaches.

When the study area gets larger, for instance at regional or country level, crop modeling or forecasting becomes more challenging due to misinformation about the geolocation of the cultivated fields per crop type. Furthermore, public institutions do not usually own such data until mid-season or later. Therefore, crop-based or remote sensing-based models applied over large areas may estimate the yield per hectare (Ton/ha) but not the total yield, at least at early stages of the crop season [54]. The main limitation of the methodology exposed in this work is that the retrieval of NDVI comes from the entire irrigated or rain-fed crop area at municipal level, not discerning potato fields (not available information), but using mean NDVI and lai values to assess the eco-climatic conditions for each date and municipality, in addition to climatic data. Similar approaches were seen in the literature for other crops [55,56]. The main advantage of remote sensing-based models is the capacity to allocate spatial information with respect to the typical crop growth models [57]. Nevertheless, the strength of the relation between remote sensing data and production forecasts is not universal and varies geographically [58]. Currently, there are some efforts to provide independent and timely information on crop areas and yields using satellite imagery. The Joint Research Center – European commission has been running a program since 1988 named as Monitoring of Agriculture using Remote Sensing (MARS) to use satellite data as well as crop-based models to provide independent and timely information on crop areas and crop yield at large scale. The performance of the MARS-crop yield forecasting system for the European Union during 1993–2015 reported a median error value across the member states of 6.3% for potato crops, although for some individual countries this error rose up to 16.69% (no $R^2$ available) [54]. The results obtained in our study ($R^2$ = 0.757/0.858, %RMSE = 18.9/14.9) are in accordance with the aforementioned errors presented in other works. This type of regional estimate of crop yield is desirable for managing large agricultural areas and determining food pricing and trading policies [59,60].

## 5. Conclusions

An Earth observation driven approach has been applied to estimate potato yield at municipal level in Mexico. The predictors were created based on, previous yield, meteorological variables, NDVI,

and farmer system information. We evaluated five machine learning algorithms (rf, svmL svmP, svmR, and glm) and six different time scenarios to carry out model predictions. The best model performance was obtained by the svmP algorithm using variables from the first five months after sowing for summer cycle ($R^2$ = 0.858 and %RMSE = 14.9), and rf algorithm using variables from the first three months after sowing for winter cycle ($R^2$ = 0.757 and %RMSE = 18.9). However, the last few months did not considerably add key information to the models. The most important variables were the ybYield and NDVI variables in the winter cycle, whereas ybYield, lailv, and laihv were the most relevant in the summer cycle. These results are supported by the large dataset used to calibrate and validated the models; and the extent of the study area, which comprises a wide range of meteorological and soil patterns. The proposed methodology can predict potato yield before harvest, which can be of great interest to establish food security strategies.

**Author Contributions:** Conceptualization, Pablo Salvador; Methodology, Pablo Salvador and Diego Gómez; Formal Analysis, Pablo Salvador; Validation, Diego Gómez, and Pablo Salvador; Data Curation, Pablo Salvador and Julia Sanz; Writing—original draft preparation, Pablo Salvador; Writing—review and editing, Diego Gómez and Pablo Salvador; Supervision, Julia Sanz and José Luis Casanova; Project Administration, José Luis Casanova. All authors have read and agreed to the published version of the manuscript.

**Funding:** This research received no external funding.

**Acknowledgments:** We acknowledge the use of data products or imagery from the Land, Atmosphere Near real-time Capability for EOS (LANCE) system operated by NASA's Earth Science Data and Information System (ESDIS) with funding provided by NASA Headquarters.

**Conflicts of Interest:** The authors declare no conflict of interest.

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
