# Peer review of "Estimation of Potato Yield Using Satellite Data at a Municipal Level: A Machine Learning Approach"

_ijgi, doi:10.3390/ijgi9060343_

Round 1

Reviewer 1 Report

The authors have made several improvements to the paper that increase its fidelity and potential impact.  However, there are some additional refinements that need to be made.  My first two comments below are the highest importance.

1) The revised paper describes an improved methodology for the study as a second/supplemental experiment.  Instead, I suggest that this should replace the random-split experiment entirely, since those results cannot be used to understand how the models would perform in a real setting.  (It is also problematic that only the best model based on the random-split experiment was used in the operational experiment.)

In formulating this experimental methodology, it is necessary to consider how the models would be used.  If all possible municipalities are already in this data set, then a partition spatially is not needed, only a partition temporally.  That is, if operationally the models need only predict yields for these same locations, then it is fine to train on the past and test on future years (like 2017 and 2018 as done in the second experiment).  If however there is a desire to deploy the trained model on new municipalities, then there would be a need to spatially partition the train and test sets.  If only temporal splitting is needed, it is possible that the variability of the results will improve.

Please update Table 3 with results for all models and use it to replace Table 2.  This experiment would also update the bar plots of feature importance in Figures 2 and 3 (since they might change). 

2) The authors did not address my request to connect the observed performance values to what is needed in a real setting.  They responded that the numbers have increased (now that they added ybYield as a variable).  I repeat my request to provide context and interpret the observed performance for its intended use case.  Are the observed R^2 and RMSE values acceptable by the intended users?

3) To explain my rating of "originality/novelty" as "low", the proposed work has applied existing machine learning methods without any change or innovation.  That is not necessarily a problem or a reason to preclude publication.  The novelty lies in the chosen problem to be solved (predicting potato yield).

Interpretation of results:

4) (I comment here on Table 2, but as above it should be replaced with Table 3; it is my hope that my comments here would still be useful with the new table.) The updated results show higher numbers than before, due to the addition of the "ybYield" variable, as we would expect.  The baseline result of simply using the ybYield variable is mentioned on lines 181-182.  Please elaborate about how this was calculated (e.g., state that it was used to predict that the current yield is unchanged from the previous year's yield, if that is what was done).  Please use the same units for RMSE as in Table 2 (e.g., 25.0% should be 0.25).  Then, follow this report with a discussion/interpretation of how the machine learning results compare (e.g., all models achieved higher R2 and lower RMSE than the baseline).

5) This statement on line 180 needs to be amended since it omits the ybYield variable (line 178) which dwarfs the importance of the NDVI variables:

"For the summer cycle, the NDVI1, NDVI2 and NDVI3 are the most important variables."

Simply stating "next most important" would not sufficiently capture the dramatic difference in importance.  I encourage the authors to describe the results as they appear in the bar plots.

6) Discuss the results in Figure 5.  (These also should be updated to instead show the results on 2017+2018 test set.)  Is there any commonality in the yields that are over-predicted?  Is there any commonality in those that are under-predicted?  Can this lead to additional improvements for those areas?

7) Table 3 should be accompanied by the same evaluation using the baseline method (predict same yield as previous year).  Otherwise it is not possible to understand if the reported (e.g., mean) R2 and RMSE are good or bad.  The discussion in lines 236-241 reports the large variation in results, which is concerning for any adoption of this method operational.  Again, is there any pattern that can be leveraged - were the results in 2017 better/worse than for 2018?  Were there some locations that had good results versus others with bad results?

8) Figure 2 showing an example of the data time series is quite interesting, but since some colors are used multiple times it is difficult to understand.  Please use different colors (or markers) for each variable.  Some additional discussion of this Figure would add insight to the paper.  For example, it seems that some variables (the flat ones) may not be informative at all for crop yield prediction. Or are they?  It also seems that the values have been normalized in some way (since they all range from 0 to 0.7 which does not seem to be a natural range for all of them given the units in Table 1.  I still do not know what "Times one" means for "total cloud cover").  Please describe all variable pre-processing and normalization in the paper.  Then, later when discussing variable importances, refer back to this Figure as appropriate.

9) I do not think the authors understood my suggestion about existing work on multiple-instance regression.  The multiple instances are the multiple pixels associated with each municipality.  Currently this work uses (if I understand correctly) the maximum NDVI value across the municipality.  Instead, one can usually get better results using all values.  It would be good to at least mention and cite this approach in the context of the current work.

10) Clarifications:

- On lines 169-170, what is "mtry"?

- Please report, for each algorithm, what parameter settings were selected as the result of the bootstrapping process.

- My suggestion to include "error bars for the 5 trials" means showing the variation in the reported R2 and RMSE results in Table 2.  I assume these are averages across the 5 trials, so they can be accompanied by how large the standard deviation was, as an "error bar".  The same applies to Table 3 (more common than showing min and max).

- I don't understand why scaling the variable importance values to 100 would improve visibility.  If the numbers were between 0 and 1 before, then this just changes the y axis, not the relative size of the bars. Please explain further what transformations were applied to the importances, or don't transform if not needed.

- Variable importance: note that the fact that a variable has a higher importance does not imply that only using one variable vs. another will result in better performance.  Line 226-227 claims "Despite NDVI was not retrieved at field level, model performances were better using this vegetation index than using only meteorological variables alone." This claim would need to be tested with an experiment to find out if it is true.  Please do this experiment, or amend the sentence to state that the current results show a higher variable importance for NDVI than for the meteorological variables, for summer crops only (which is what is currently demonstrated).

- Figure 5 caption: add note that this is the test set (as indicated in the text) and swap the axes so that actual yield is the independent variable (x axis) and predicted yield is the y axis.  That allows us to easily see where the model predicts too high or too low.

- Table 3 caption: add that this is the simulation of an operational test on 2 held-out years and XXX held-out locations (I am not sure what XXX is).

- Please carefully read through the new text that has been added; in some cases the grammar and capitalization needs improvement.

Thank you for your continued efforts on this paper and this experimental area!

Author Response

We would like to thank Reviewer 1 for his/her comments and constructive suggestions concerning this manuscript. We believe that the quality of the manuscript has improved and that the results obtained are far more conclusive than the former ones. All the changes have been clearly highlighted in the original manuscript with the Microsoft Word “track changes” option.

The authors have made several improvements to the paper that increase its fidelity and potential impact.  However, there are some additional refinements that need to be made.  My first two comments below are the highest importance.

1) The revised paper describes an improved methodology for the study as a second/supplemental experiment.  Instead, I suggest that this should replace the random-split experiment entirely, since those results cannot be used to understand how the models would perform in a real setting.  (It is also problematic that only the best model based on the random-split experiment was used in the operational experiment.)

In formulating this experimental methodology, it is necessary to consider how the models would be used.  If all possible municipalities are already in this data set, then a partition spatially is not needed, only a partition temporally.  That is, if operationally the models need only predict yields for these same locations, then it is fine to train on the past and test on future years (like 2017 and 2018 as done in the second experiment).  If however there is a desire to deploy the trained model on new municipalities, then there would be a need to spatially partition the train and test sets.  If only temporal splitting is needed, it is possible that the variability of the results will improve.

Thank you for this comment. We agree with Reviewer 1 in his/her comment. For this reason, we have changed the methodology, using data before 2016 as training-validation and data of 2017 and 2018 as the independent test. Models were trained 50 times with random selection of the 90% of each data set (train and test).

Please update Table 3 with results for all models and use it to replace Table 2.  This experiment would also update the bar plots of feature importance in Figures 2 and 3 (since they might change). 

Thank you for the comment. In the current version, Table 2 shows the model performance per algorithm, cycle and months used to create variables from the harvest date using the bootstrapping method (25 repetitions) and 1 iteration. Table 3, instead, shows the model performance of each algorithm, cycle and months (used to create variables from the harvest date) across 50 iterations (not taking 10 % of the dataset at each iteration), expressed as mean values of R2 and RMSE, and their corresponding standard deviations. With Table 3, we want to ensure that model results presented low variance.

We think that joining these two tables would result in a too big table, hence less readable than if divided into two.

Yes, bar plots have change and both have been replaced with new ones.

2) The authors did not address my request to connect the observed performance values to what is needed in a real setting.  They responded that the numbers have increased (now that they added ybYield as a variable).  I repeat my request to provide context and interpret the observed performance for its intended use case.  Are the observed R^2 and RMSE values acceptable by the intended users?

The answer to that question depends on the expectations of the final user, which surely varies across organizations, countries, etc. In the revised version, we have compared model performances with respect to a baseline requirement (as has been suggested by the reviewer) that consist on taking the yield production during the year before as predictor. The obtained results when using the latter model were R2 = 0.515 and %RMSE = 25.0 (winter cycle) and R2 = 0.766 and %RMSE = 20.5 (summer cycle). From our point of view, the results presented in this study represents an advance with respect to the base-line approach (the yield of the year before) and we have proved that yield prediction can be improved with other variables retrieved by remote sensing techniques.

3) To explain my rating of "originality/novelty" as "low", the proposed work has applied existing machine learning methods without any change or innovation.  That is not necessarily a problem or a reason to preclude publication.  The novelty lies in the chosen problem to be solved (predicting potato yield).

Interpretation of results:

4) (I comment here on Table 2, but as above it should be replaced with Table 3; it is my hope that my comments here would still be useful with the new table.) The updated results show higher numbers than before, due to the addition of the "ybYield" variable, as we would expect.  The baseline result of simply using the ybYield variable is mentioned on lines 181-182.  Please elaborate about how this was calculated (e.g., state that it was used to predict that the current yield is unchanged from the previous year's yield, if that is what was done).  Please use the same units for RMSE as in Table 2 (e.g., 25.0% should be 0.25).  Then, follow this report with a discussion/interpretation of how the machine learning results compare (e.g., all models achieved higher R2 and lower RMSE than the baseline).

The rationale behind the use of ybYield as variable has been included in section 2.3.1.

RMSE units have been homogenized.

A brief comment has been added to show that better results were found using ML algorithms with remotely sensed variables such as NDVI or meteorological data.

5) This statement on line 180 needs to be amended since it omits the ybYield variable (line 178) which dwarfs the importance of the NDVI variables:

"For the summer cycle, the NDVI1, NDVI2 and NDVI3 are the most important variables."

Simply stating "next most important" would not sufficiently capture the dramatic difference in importance.  I encourage the authors to describe the results as they appear in the bar plots.

The results, discussion and conclusion section have been extensively modified as new results have been included in the revised version of the manuscript. In addition, we added further comments about the importance of YbYield as critical variable in model performances. 

6) Discuss the results in Figure 5.  (These also should be updated to instead show the results on 2017+2018 test set.)  Is there any commonality in the yields that are over-predicted?  Is there any commonality in those that are under-predicted?  Can this lead to additional improvements for those areas?

Figure has been replaced by a new one (based on the new test set) where additional information has been added to the figure (ML algorithm, irrigation vs rain-fed).

The over/under predicted values do not seem to follow any pattern and are not related to the year (2017 or 2018). Neither are related to the farmer system or ybYield.

7) Table 3 should be accompanied by the same evaluation using the baseline method (predict same yield as previous year).  Otherwise it is not possible to understand if the reported (e.g., mean) R2 and RMSE are good or bad.  The discussion in lines 236-241 reports the large variation in results, which is concerning for any adoption of this method operational.  Again, is there any pattern that can be leveraged - were the results in 2017 better/worse than for 2018?  Were there some locations that had good results versus others with bad results?

The large variations seen in the results have disappeared in the new approach. We consider that this is due to the number of samples present in the test and train sets. As commented before, we do not see any spatial or temporal pattern.

8) Figure 2 showing an example of the data time series is quite interesting, but since some colors are used multiple times it is difficult to understand.  Please use different colors (or markers) for each variable.  Some additional discussion of this Figure would add insight to the paper.  For example, it seems that some variables (the flat ones) may not be informative at all for crop yield prediction. Or are they?  It also seems that the values have been normalized in some way (since they all range from 0 to 0.7 which does not seem to be a natural range for all of them given the units in Table 1.  I still do not know what "Times one" means for "total cloud cover").  Please describe all variable pre-processing and normalization in the paper.  Then, later when discussing variable importances, refer back to this Figure as appropriate.

Markers have been added improving visibility, thanks for the comment. The values have been normalized, it was visible in the figure legend but now it is highlighted in the text for the sake of clarity.

Times one is just percent divided by 100, we found the expression in several works, but we can replace it by "%/100".

Importance variables have been discussed in the discussion section, line 308.

9) I do not think the authors understood my suggestion about existing work on multiple-instance regression.  The multiple instances are the multiple pixels associated with each municipality.  Currently this work uses (if I understand correctly) the maximum NDVI value across the municipality.  Instead, one can usually get better results using all values.  It would be good to at least mention and cite this approach in the context of the current work.

Reviewer 1 is right, and it was not well explained in the text. The maximum value within each municipality was not taken as the representative value. For each pixel, we computed the maximum value of the month (for some months, there is one value while for others, there are two) and then, we calculate the mean value for all the pixels classified as crop within the municipality. The paragraph has been rewritten in order to better explain the method.

10) Clarifications:

- On lines 169-170, what is "mtry"?

Sorry, it has been included in the text, "number of variables tried at each split".

- Please report, for each algorithm, what parameter settings were selected as the result of the bootstrapping process.

We have included in the text which were the best hyper-parameters for rf and svmP model after the 50 iterations.

- My suggestion to include "error bars for the 5 trials" means showing the variation in the reported R2 and RMSE results in Table 2.  I assume these are averages across the 5 trials, so they can be accompanied by how large the standard deviation was, as an "error bar".  The same applies to Table 3 (more common than showing min and max).

Thank you for the suggestion. Table 3 shows the model results over 50 iterations. In order to show how variable the results were, we changed the previously presented max. and min. scores by the standard deviations of the mean RMSE and R2. Thus, we consider that we address the point raised by the reviewer. Since Table 2 presents model performances of only one iteration,it does not indicate any measure of dispersion.

- I don't understand why scaling the variable importance values to 100 would improve visibility.  If the numbers were between 0 and 1 before, then this just changes the y axis, not the relative size of the bars. Please explain further what transformations were applied to the importances, or don't transform if not needed.

The scaling is made by the varImp function mentioned in the text. If it is not scaled, the number is not between 0 and 1 making the comparison difficult as only the first 20 variables are retrieved.

- Variable importance: note that the fact that a variable has a higher importance does not imply that only using one variable vs. another will result in better performance.  Line 226-227 claims "Despite NDVI was not retrieved at field level, model performances were better using this vegetation index than using only meteorological variables alone." This claim would need to be tested with an experiment to find out if it is true.  Please do this experiment, or amend the sentence to state that the current results show a higher variable importance for NDVI than for the meteorological variables, for summer crops only (which is what is currently demonstrated).

The sentence has been rewritten since only in summer we observe that NDVI had a greater influence in model results.

- Figure 5 caption: add note that this is the test set (as indicated in the text) and swap the axes so that actual yield is the independent variable (x axis) and predicted yield is the y axis.  That allows us to easily see where the model predicts too high or too low.

It has been done.

- Table 3 caption: add that this is the simulation of an operational test on 2 held-out years and XXX held-out locations (I am not sure what XXX is).

Thank you for the suggestion. We have added that these results correspond to the model performance over the test dataset (2017 and 2018) over 50 iterations.

- Please carefully read through the new text that has been added; in some cases the grammar and capitalization needs improvement.

Thank you for your continued efforts on this paper and this experimental area!

Thanks to reviewer comments, we consider that the revised version of the manuscript is much more complete.

Submission Date

30 March 2020

Date of this review

18 Apr 2020 21:22:43

Reviewer 2 Report

The paper discusses the critical domain of EO driven analysis for reliable, accurate and timely crop yield forecasts. Unfortunately, the referencing is not up-to-date. Please take into account up-to date reviews (doi.org/10.1016/j.agsy.2018.05.010).

Line 39: please rephrase destination with a more appropriate word

Lines 51-64 Please specify clearly the problem you perceive, present up-to date approaches, and avoid a simple presentation of Vegetation indices. The introduction is not the appropriate section to present the NDVI formula, move it to materials and methods. Maybe references to recent review and works (as indicated above) will be helpful in order to enhance your statement.

Line 54 Delete 10,11 you have already mentioned the relevant works at the beginning of this sentence.

Line 56...the fraction...

Line 71 In my view you should re-write your objective: The overarching objective of the current research work is to evaluate the performance of well-established non-linear machine learning algorithms to estimate the potato yield prior the harvesting period…

Line 85 Please make subsection for the following paragraphs (etc. In situ data, Remote sesing data etc.)

Line 130 Please provide a section that describes the creation of temporal patterns and a new one that describes the machine learning modelling

Line 136-138 There are no differences in phenological stages that may influence the cropping cycles?

Line 139 Usually numbers up to ten are generally written in full text

Line 143-145 Please remove this sentence. Maybe you can move it in the introduction.

Line 159 Is this true? The results and their explanation are not clear. Looking the Table 1 we observed higher values in earlier stages. For instance looking rf model performance can we conclude that we can predict potato wield with the same performance using 1 and 6 month data, accordingly?

Line 174-183 In a previous work (doi.org/10.3390/rs11151745) the authors evaluate the performance of more Vis. Could you consider the inclusion of more indices to extract better insights? Moreover, what is the performance of the proposed model without the inclusion of ybYield? It could be valuable to have these results to assess the limitations whereby standalone remote sensing is not sufficient to reach the desired accuracy.

Line 188 What about the differences of irrigated and non-irrigated. Could you present and visualize these results?

Line 253-272 Data recorded by EO sensors cannot ideally provide production forecasts as ideally needed for food security interventions. Please provide a more critical review here.

Author Response

We would like to thank Reviewer 2 for his/her comments and constructive suggestions concerning this manuscript. We believe that the quality of the manuscript has improved and that the results obtained are far more conclusive than the former ones. All the changes have been clearly highlighted in the original manuscript with the Microsoft Word “track changes” option.

The paper discusses the critical domain of EO driven analysis for reliable, accurate and timely crop yield forecasts. Unfortunately, the referencing is not up-to-date. Please take into account up-to date reviews (doi.org/10.1016/j.agsy.2018.05.010).

Line 39: please rephrase destination with a more appropriate word

Destination has been replaced by "end use".

Lines 51-64 Please specify clearly the problem you perceive, present up-to date approaches, and avoid a simple presentation of Vegetation indices. The introduction is not the appropriate section to present the NDVI formula, move it to materials and methods. Maybe references to recent review and works (as indicated above) will be helpful in order to enhance your statement.

Thank you for the suggestion. To better contextualize the problem we aim to address (crop yield prediction using non-field data), we have added a reference to a recent article in which they used a vegetation index (NDVI) to assess the general crop status of an area (without distinguishing among crops) to predict soybean.

We also agree with the Reviewer that the introduction is not the right place to introduce the NDVI formula, hence we moved it into the Materials and Methods section."

Line 54 Delete 10,11 you have already mentioned the relevant works at the beginning of this sentence.

Yes, you are right it is a mistake, 10 and 11 references are deleted from the text.

Line 56...the fraction...

Done

Line 71 In my view you should re-write your objective: The overarching objective of the current research work is to evaluate the performance of well-established non-linear machine learning algorithms to estimate the potato yield prior the harvesting period…

The objective has been re-written based on the comment provided by Reviewer.

Line 85 Please make subsection for the following paragraphs (etc. In situ data, Remote sensing data etc.)

Thank you for the advice. We have made subsections as suggested by the Reviewer.

Line 130 Please provide a section that describes the creation of temporal patterns and a new one that describes the machine learning modelling

Thanks for the advice, we have included in the revised version of the manuscript.

Line 136-138 There are no differences in phenological stages that may influence the cropping cycles?

There may certainly be some differences in the phenological stages of potato crops across municipalities and states given the large extension of Mexico. Therefore, it should be expected that those stages may also have some influence in the cropping cycles. In this study, we do not have enough information to address and/or evaluate those influences addressed by Reviewer 2 at field scale. Nevertheless, we used the cropping cycle to classify both groups so that the models, inherently, associate the crop cycle to the rest of variables of the municipality (NDVI, meteorological variables, etc.)

Line 139 Usually numbers up to ten are generally written in full text

Numbers have been replaced by full text.

Line 143-145 Please remove this sentence. Maybe you can move it in the introduction.

Thanks for the comment, the sentence has been moved into the introduction section.

Line 159 Is this true? The results and their explanation are not clear. Looking the Table 1 we observed higher values in earlier stages. For instance looking rf model performance can we conclude that we can predict potato wield with the same performance using 1 and 6 month data, accordingly?

Results, discussion and conclusion sections have been extensively modified since new results have been included in the manuscript. Reviewer 2 is right considering that model performances were quite similar taking less or more months. As discussed in the manuscript, we explain that the ybYield variable was of great importance for the winter cycle to complement information provided by the NDVI in the first two months. In the summer cycle, the variable importance analysis indicated other predictors, instead, such as lailv, laihv, NDVI4 and NDVI5.

Line 174-183 In a previous work (doi.org/10.3390/rs11151745) the authors evaluate the performance of more Vis. Could you consider the inclusion of more indices to extract better insights? Moreover, what is the performance of the proposed model without the inclusion of ybYield? It could be valuable to have these results to assess the limitations whereby standalone remote sensing is not sufficient to reach the desired accuracy.

We evaluated the possibility to include more vegetation indices with MODIS data, but given the coarse spatial resolution (500m and 1000m) of this sensor and the lack of data at field-scale represent major problems when we need to evaluate over the crop land cover. The behavior of the reflectance and meteorological variables perform very different in this context.

The evaluation of the RMSE score in the final models without the inclusion of ybYield has been undertaken. We present this information in the results and discussion sections, where we highlight the need to include this variable into our models to reach better predictions.

Line 188 What about the differences of irrigated and non-irrigated. Could you present and visualize these results?

Irrigated and non irrigated results have been presented on lines 213-215. In Figure 5, we used different colors to distinguish them.

Line 253-272 Data recorded by EO sensors cannot ideally provide production forecasts as ideally needed for food security interventions. Please provide a more critical review here.

Reference has been added in discussion section lines 362-365 in order to review the problems in the task to establish relations between remote sensing data and yield production.

Meroni, M.; Rembold, F.; Verstraete, M. M.; Gommes, R.; Schucknecht, A.;  Beye, G. Investigating the relationship between the inter-annual variability of satellite-derived vegetation phenology and a proxy of biomass production in the Sahel. Remote Sensing 2014, 6(6), 5868-5884.

Submission Date

30 March 2020

Date of this review

02 Apr 2020 21:20:56

Round 2

Reviewer 2 Report

Overall comment

The authors responded to all the comments. I appreciate that they provided additional results (e.g. non-inclusion of ybYield). In the first round I asked to evaluate the performance of more VIS. The authors in their response indicated that the coarse resolution and the lack of data at field scale are the major limitations without any explanation. I am not satisfied with their reply. It would have been very useful if the authors had explained how other VIS influence the behavior of the reflectance and meteorological variables as they mentioned. The discussion section is very poor and very general. The limitation of this study should be discussed in more detail, e.g. lack of in situ data for phenological information, inclusion of other VIS.

Other specific comments:

Line 11 In my view it is better to said ...provided by...

Line 90 climatology refers to the study of climate, in my view climatic conditions is a more appropriate word

Line 127 the laws of physics, what do you mean, also what is the number 31? Please rephrase.

Line 154 how many observations?

Line 169-171 Usually numbers up to ten are generally written in full text

 Line 175-176 The authors should explained why they change the data splitting to 85%-15%. In my view the val dataset is really small to generalize the results. This is considered as a major issue since the authors change their methodology.

Line 384 maybe an Earth Observation driven approach, instead of A machine learning approach

Author Response

We thank Reviewer 2 for his/her helpful and constructive comments.  

Overall comment

The authors responded to all the comments. I appreciate that they provided additional results (e.g. non-inclusion of ybYield). In the first round I asked to evaluate the performance of more VIS. The authors in their response indicated that the coarse resolution and the lack of data at field scale are the major limitations without any explanation. I am not satisfied with their reply. It would have been very useful if the authors had explained how other VIS influence the behavior of the reflectance and meteorological variables as they mentioned. The discussion section is very poor and very general. The limitation of this study should be discussed in more detail, e.g. lack of in situ data for phenological information, inclusion of other VIS.

The use of other VIS (with the same spatial resolution as NDVI) such as EVI were dismissed since they generally provide related information (Wardlow  et al., 2007;Wardlow & Egbert, 2010). Other VIS derived from MODIS have coarser spatial resolution (> 250 m) (Teillet et al., 1997) and/or are not aggregated in a 16-day product (which is done by NASA with their own protocols).

Phenological information (and specially sowing and harvest dates) per municipality would have been an important source of information. Unfortunately, such information is not contained in the database provided by SIAP (Servicio de Información Agroalimentaria y Pesquera, Government of Mexico).

All these issues have been included in the revised version of the manuscript in the materials and discussion sections.

Wardlow, B.D.; Egbert, S.L.; Kastens, J.H. Analysis of time-series MODIS 250 m vegetation index data for crop classificaation in the U.S. Central Great Plains. Remote Sensing of Environment 2007, 108, 290-310.

Wardlow, B. D., & Egbert, S. L. (2010). A comparison of MODIS 250-m EVI and NDVI data for crop mapping: a case study for southwest Kansas. International Journal of Remote Sensing, 31(3), 805-830.

Teillet, P.M.; Staenz, K.; William, D.J. Effects of spectral, spatial, and radiometric characteristics on remote sensing vegetation indices of forested regions. Remote Sensing of Environment 1997, 61(1), 139-149

Other specific comments:

Line 11 In my view it is better to said ...provided by...

The sentence has been replaced according to the reviewer comment.

Line 90 climatology refers to the study of climate, in my view climatic conditions is a more appropriate word

The reviewer is right; climatology has been replaced.

Line 127 the laws of physics, what do you mean, also what is the number 31? Please rephrase.

The “31” should have been in the reference´s format. In this reference, readers can find the details about ERA5 datasets and processing. The sentence has been rewritten including more details about ERA5 (model physics, core dynamics and data assimilation).

In the current version, it is the [34] since more reference has been added to the manuscript.

Line 154 how many observations?

The MOD13Q1 product is a 16-day composite. Thus, there were some months with only one NDVI image and others with two NDVI images. In the latter case (two NDVI images whose dates coincide in the same month), we chose those values of NDVI (by pixels) which were the highest. We have further clarified it in the text.

Line 169-171 Usually numbers up to ten are generally written in full text

Our apologies because we overlooked this comment provided in the previous revision of the manuscript. We thought that all the numbers (up to ten) had been re-written in full text format. We have modified it.

 Line 175-176 The authors should explained why they change the data splitting to 85%-15%. In my view the val dataset is really small to generalize the results. This is considered as a major issue since the authors change their methodology.

To test our models with independent data (not used for calibration or optimization), we were suggested by one of the reviewers to use the last years of the original dataset. Thus, we would simulate the performance of the models in a more “real-world scenario”. We needed to find a trade-off between the number of hold-out samples in the test set and at the same time, keep the most recent data to test the models. With this method, we tested the models for 126  samples which correspond to the last two years of our dataset (2017 and 2018).

We have included additional information to the manuscript to reinforce this point.

Line 384 maybe an Earth Observation driven approach, instead of A machine learning approach

We have rewritten the sentence according to the reviewer´s comment.

Submission Date

30 March 2020

Date of this review

04 May 2020 23:09:56

This manuscript is a resubmission of an earlier submission. The following is a list of the peer review reports and author responses from that submission.

Round 1

Reviewer 1 Report

The manuscript discusses the necessity of timely and accurate forecasts of agricultural production at national scale through the use of Earth Observations. However, the current study strongly builds on the implementation of common machine learning methods and the novel or original part of this study needs to be brought out clearly. Now it seems that there is a lack of novelty. For instance, you have to propose a new ML method and then compare your results with the methods listed in the current version. Moreover, for future submissions, I suggest evaluating the potential changes in the phenological patterns. Is there any significant difference among the years that may influence the growing period? Maybe a comparative analysis among Vis can provide also useful insights for further studies.

Reviewer 2 Report

This paper focuses on an important and valuable task, of estimating potato yield prior to harvest using machine learning applied to remote sensing and meterological data.  The main contribution is the application and evaluation of existing classical machine learning methods rather than the invention of new approaches.

The introduction provides very good motivation for the work.  The primary limitation of the work is in the methodology.  Several important issues and aspects of the data are not considered, which affects the reliability of the evaluation (how well it could generalize to new data in the future and be used by others).  It is likely that performance is over-estimated due to correlations in spatial and temporal dimensions.  Namely:

1) The data is spatially correlated, so randomly splitting into train/test sets allows correlation between train and test sets. Instead, the data should be split spatially to better estimate how the models will generalize to new locations.

2) The evaluation is not organized according to the temporal nature of the data.  Models should be trained on past data and used to predict future data, instead of mixing all years together.  This would better simulate how the models would be used in reality.

3) Section 2.3 lists 5 machine learning algorithms but does not specify what hyperparameter settings were used, or if/how they were optimized for this data set.  This makes the work unreproducible.  It is also possible that better performance would be achieved after optimizing the hyperparameters (if this was not done).

4) There is no comparison to a simple baseline/control (e.g., for each location, predict that the yield this year will be the same as the yield last year, using no input variables from the current year).  In addition, there is no empirical comparison to other methods from literature (refs 21-24, 49) on the same data set.

5) The results are poorly and selectively interpreted.  The discussion emphasizes a minor difference in R^2 but does not comment on the fact that there is effectively no difference in RMSE for any method, in either cycle. What does this mean?  Why was it ignored?

No error bars for the 5 trials are provided.

The paper claims that the current results "are in accordance" (line 222) with other studies, but the numeric results seem much worse than other studies, and regardless it is not clear that they are comparable (not on the same data, same data types, same spatial granularity (city vs. state), and not with same methodology).

6) There is no explanation for how variable importance was calculated.

7) There is a claim (line 187) that performance was better when using NDVI than only meteorological inputs, but this is not substantiated in the paper.

8) There has been a lot of work on this problem that approaches it as a multiple-instance regression (MIR) task, since the granularity of crop yield measurements is usually much larger than the remote sensing data (i.e., multiple remote sensing pixels cover the area for which the yield is measured).  An MIR approach enables the inclusion of multiple remote sensing observations per target instead of summarizing them with (in the paper under review) the maximum value observed.  The current study would be improved by trying out these methods as well. 

One place to start would be "Mixture Model for Multiple Instance Regression and Applications in Remote Sensing" by Wang et al., 2011:

  https://ieeexplore.ieee.org/abstract/document/6088008

9) Even given the likelihood that performance is over-estimated, the numbers still do not seem very good.  There is a final missing link connecting these numbers to those who would use them.  Is an RMSE of 0.24 acceptable?  Is an R^2 of 0.6 or 0.7 good enough?  The paper should analyze and interpret these results from the context of the end user, not solely compare to other studies done in different settings.

Minor suggestions:

  • It would help to define and show the equation for NDVI the first time it is mentioned.
  • Section 2.2 would be enriched by showing an example plot of one location's time series (MODIS and meterological data).
  • Units in Table 1 should be explained (what is M? what is "Times one")) and exponents shown correctly (m^2 instead of m2).
  • Section 2.3: whose implementation of the ML algorithms was used? Include a citation.
  • Fig 2: variable importance should be a bar plot or table, not a continuous line plot.
  • Figure 4 is never discussed in the text.
  • Author contributions are not filled in (generic template text is used).